# Maternal Immune Activation Causes Social Behavior Deficits and Hypomyelination in Male Rat Offspring with an Autism-Like Microbiota Profile

**DOI:** 10.3390/brainsci11081085

**Published:** 2021-08-18

**Authors:** Gilbert Aaron Lee, Yen-Kuang Lin, Jing-Huei Lai, Yu-Chun Lo, Yu-Chen S. H. Yang, Syuan-You Ye, Chia-Jung Lee, Ching-Chiung Wang, Yung-Hsiao Chiang, Sung-Hui Tseng

**Affiliations:** 1Department of Medical Research, Taipei Medical University Hospital, Taipei 110, Taiwan; gilbertlee@tmu.edu.tw (G.A.L.); yoyoyei330@gmail.com (S.-Y.Y.); 2Department of Microbiology and Immunology, School of Medicine, College of Medicine, Taipei Medical University, Taipei 110, Taiwan; 3Graduate Institute of Athletics and Coaching Science, National Taiwan Sport University, Taoyuan 333, Taiwan; robbinlin@ntsu.edu.tw; 4Core Laboratory of Neuroscience, Office of R&D, Taipei Medical University, Taipei 110, Taiwan; M105095006@tmu.edu.tw (J.-H.L.); ychiang@tmu.edu.tw (Y.-H.C.); 5Center for Neurotrauma and Neuroregeneration, Taipei Medical University, Taipei 110, Taiwan; 6PhD Program for Neural Regenerative Medicine, College of Medical Science and Technology, Taipei Medical University, Taipei 110, Taiwan; aricalo@tmu.edu.tw; 7Joint Biobank, Office of Human Research, Taipei Medical University, Taipei 110, Taiwan; can_0131@tmu.edu.tw; 8PhD Program for Clinical Drug Discovery of Herbal Medicine, College of Pharmacy, Taipei Medical University, Taipei 110, Taiwan; cjlee@tmu.edu.tw; 9Graduate Institute of Pharmacognosy Science, College of Pharmacy, Taipei Medical University, Taipei 110, Taiwan; 10Department of Pharmaceutical Sciences, Taipei Medical University, Taipei 110, Taiwan; crystal@tmu.edu.tw; 11Department of Surgery, College of Medicine, Taipei Medical University, Taipei 110, Taiwan; 12Graduate Institute of Neural Regenerative Medicine, College of Medical Science and Technology, Taipei Medical University, Taipei 110, Taiwan; 13Department of Physical Medicine and Rehabilitation, School of Medicine, College of Medicine, Taipei Medical University, No. 250 Wu Hsing Street, Taipei 110, Taiwan; 14Department of Physical Medicine and Rehabilitation, Taipei Medical University Hospital, Taipei 110, Taiwan

**Keywords:** autism spectrum disorder, lipopolysaccharide, maternal immune activation, microbiota, three-chamber test, myelination, brain–gut–microbiota axis, social behavior deficits

## Abstract

Maternal immune activation (MIA) increases the risk of autism spectrum disorder (ASD) in offspring. Microbial dysbiosis is associated with ASD symptoms. However, the alterations in the brain–gut–microbiota axis in lipopolysaccharide (LPS)-induced MIA offspring remain unclear. Here, we examined the social behavior, anxiety-like and repetitive behavior, microbiota profile, and myelination levels in LPS-induced MIA rat offspring. Compared with control offspring, MIA male rat offspring spent less time in an active social interaction with stranger rats, displayed more anxiety-like and repetitive behavior, and had more hypomyelination in the prefrontal cortex and thalamic nucleus. A fecal microbiota analysis revealed that MIA offspring had a higher abundance of *Alistipes*, *Fusobacterium*, and *Ruminococcus* and a lower abundance of *Coprococcus*, *Erysipelotrichaies*, and *Actinobacteria* than control offspring, which is consistent with that of humans with ASD. The least absolute shrinkage and selection operator (LASSO) method was applied to determine the relative importance of the microbiota, which indicated that the abundance of *Alistipes* and *Actinobacteria* was the most relevant for the profile of defective social behavior, whereas *Fusobacterium* and *Coprococcus* was associated with anxiety-like and repetitive behavior. In summary, LPS-induced MIA offspring showed an abnormal brain–gut–microbiota axis with social behavior deficits, anxiety-like and repetitive behavior, hypomyelination, and an ASD-like microbiota profile.

## 1. Introduction

Maternal immune activation (MIA) has been linked to an increased risk of neurodevelopmental psychiatric disorders in offspring [1,2]. Animal models of MIA have been developed by activating the immune system with immunogens during pregnancy and then observing the development of defective behaviors in offspring with autism-like behavior [3]. MIA generates inflammatory cytokines to which the fetus is exposed during mid-gestation, possibly affecting fetal brain development [4]. Furthermore, the presence of inflammatory molecules and cytokines can affect central nervous system development [5] and adversely affect neuron survival [6].

Gut microbiota plays a critical role in regulating host physiology, metabolism, nutrition, and brain function [7]. Microbial dysbiosis is correlated with various adverse consequences, including behavioral abnormalities, neuropathology, immune dysfunction, and deficient gastrointestinal integrity [7]. The prenatal environment affects the microbiome of offspring [8]. Microbial dysbiosis is associated with the symptoms of autism spectrum disorder (ASD), including impaired social communication and repetitive behaviors [9]. However, how prenatal infection affects the brain–gut–microbiota axis, which regulates behavioral phenotypes, remains unclear.

Maternal lipopolysaccharide (LPS) exposure causes reproductive, behavioral, and neurochemical abnormalities in offspring [10]. LPS, an endotoxin, activates immune cells to release proinflammatory cytokines, which induce maternal cytokine responses and may increase the risk of atypical brain development [11]. Prenatal LPS treatment causes social behavior deficits in male offspring [9,12]. Pregnant rats injected with intraperitoneal LPS on gestation day (GD) 9.5 were demonstrated to induce the most relevant long-term neuropathological consequences in offspring [13,14]. However, the microbiota changes have not been quantified in these MIA offspring.

In this study, we investigated the relationship between microbiota and ASD-related behavior and demonstrated brain myelination changes in MIA offspring. Furthermore, we used the least absolute shrinkage and selection operator (LASSO) method to determine the most relevant microbiota genera associated with ASD-related behaviors. Taken together, we demonstrated that an abnormal brain–gut–microbiota axis with phenotypes includes a social behavior deficit, anxiety-like and repetitive behavior, hypomyelination, and dysbiosis microbiota in MIA offspring; thus, providing a link between maternal infection and the etiopathogenesis of ASD.

## 2. Materials and Methods

### 2.1. LPS-Induced MIA Rat Model

All animal procedures were approved by the Animal Care and Use Committees of Taipei Medical University. Eight-week-old Wistar female rats (BioLASCO, Taipei, Taiwan) in their first pregnancy were used in this study. Female rats were mated overnight with male rats with mating experience and were checked for the presence of a vaginal plug the subsequent morning to confirm mating. Pregnant rats were housed individually and allowed to raise their own litters until weaning. On GD 9.5, 500 μg/kg LPS (*Escherichia coli* O127:B8) or phosphate-buffered saline (PBS) was injected intraperitoneally into pregnant rats. Litters were left undisturbed until weaning at postnatal day 21. Offspring were housed in same-sex cages containing three rats until the end of the experiments. All animals were housed in temperature-controlled rooms under a 12 h light/dark cycle with ad libitum access to water and same food. All behavioral testing was performed during rats’ light cycle between 9 a.m. and 5 p.m. The experiments were performed in accordance with guidelines by the International Council for Laboratory Animal Science (ICLAS) for the care and use of laboratory animals for experiments.

### 2.2. Three-Chamber Test

The three-chamber social interaction test was adapted from a previous study [15]. Stranger rats were age- and sex-matched with the testing rats. The body weights of stranger rats at 5 and 7 weeks were 130 and 195 g, respectively. Stranger rat was placed in the right-side chamber of three-chambered apparatus (Deep Brain Tech, Taiwan) to test the social interaction between stranger and the test rat during a 10 min test session (Figure 1B). The left chamber remained empty. The sociability score, used to measure the social preference of test rat was defined as the time spent in the social region of the central space near stranger rat during test session (Figure 1B). Test rat’s behavior was recorded using a camera (The Imaging Source, DFK 33UP1300, Bremen, Germany). OptiMouse software was used to calculate the total time spent by rats in three zones of the central space (nonsocial, center, and social regions) in a three-chambered apparatus (Figure 1B).

### 2.3. Marble-Burying Test

The marble-burying test was used to examine the test rat’s anxiety-like and repetitive behavior. A clean cage (22 × 45 × 20 cm^3^) was prepared with a 4 cm corncob bedding material containing 20 embedded marbles. After 30 min, the number of marbles that remained buried in the corncob bedding was recorded.

### 2.4. Novel Object Recognition Test

Recognition memory was evaluated using the novel object recognition (NOR) test, which was used to compare the amount of time a rat spent investigating a novel object versus a familiar object. The familiar object was a plastic square block, and the novel object was a plastic V-shaped block. First, rats were habituated to an open-field Plexiglas arena (60 × 60 × 100 cm^3^) for 10 min. After habituation, the rats were allowed to explore two identical familiar objects for 10 min during the familiarization phase. After a 24 h intertrial interval, one familiar object was replaced with a novel object, and the rats were returned to the arena and allowed to explore the two objects. A preference index, which measured the time spent by the test rat exploring the novel object over the familiar one, was calculated as a percentage using the following equation: ((Time N − Time F)/(Time N + Time F)) × 100, where F and N represent the time spent near the familiar and novel objects, respectively.

### 2.5. Open-Field Test

Using an open-field test, we evaluated the general motor activity and anxiety-related exploratory activity in the MIA and control rats. Rats were first habituated to an open-field Plexiglas arena (60 × 60 × 100 cm^3^) for 10 min. Their locomotor activity and anxiety-like behavior were monitored for 10 min using open-field tests and recorded using an EthoVision system. General locomotor activity was defined as the total distance traveled in the open field. Anxiety-like behavior was measured based on the number of entries and retention times in the wall zone of the open field. The center of the open field was defined as a 30 × 30-cm^2^ area in the geometric center of the arena. The wall zone was defined as a peripheral zone 5 cm from all four sides.

### 2.6. 16S rRNA Gene Sequencing and Next-Generation Sequencing

The stool samples of the rats were purified using the QIAamp Fast DNA Stool Mini Kit (QIAGEN, Germany). Library preparation was conducted following the protocol of 16S ribosomal RNA gene amplicons for the Illumina MiSeq system. Sequence reads were deposited in the European Nucleotide Archive (accession number: PRJEB28574). Universal primers (341F and 805R) were used to amplify the V3–V4 region of bacterial 16S rRNA genes; they were first removed from demultiplexed, paired reads using Cutadapt (v1.12; DOI:10.14806/ej.17.1.200). The filtered reads were then processed using the DADA2 package (v1.3.5) in R (v3.3.3) [16], following the workflow described by Callahan et al. [17], but without the rarefying procedure. Briefly, the forward and reverse reads were filtered and trimmed based on the read quality score and read length. Dereplication was then performed to merge identical reads, and the reads were then subjected to the denoising algorithm DADA2; the reads alternated between error rate estimation and sample composition inference until they converge into a consistent solution. Finally, paired reads with ≥20-bp overlap were merged, and chimeras were removed. A list of V3–V4 sequence variants found in the samples, which were inferred with DADA2, and the frequency of each sequence variant in each sample were obtained. Taxonomy assignment was performed using the SILVA database (v128) [18] for reference with a minimum bootstrap confidence of 80. Multiple sequence alignment of variants was performed using DECIPHER (v2.2.0) and the phylogenetic tree was constructed from the alignment using phangorn (v2.2.0) [19]. The count table, taxonomy assignment results, and phylogenetic tree were consolidated into a phyloseq object, and community analyses were performed using phyloseq (v1.19.1) [20]. The alpha-diversity indices were calculated using the estimated richness function of the phyloseq package. Data from the treatment and control groups were compared using the Wilcoxon–Mann–Whitney test (at α = 0.05). UniFrac distances were calculated using the GUniFrac package (v1.1) to assess community dissimilarity between the groups [21]. Principal coordinate analysis ordination on UniFrac distances was performed, and the adonis and betadisper functions from the vegan package (v2.4; https://CRAN.R-project.org/package=vegan, accessed on 24 August 2017) were used to analyze the dissimilarity of composition among the groups and the homogeneity of dispersion, respectively. Vegan is R package which provides tools for descriptive community ecology. It includes functions of diversity analysis, community ordination, and dissimilarity analysis. Adonis and Betadisper are the sister functions of Vegan package. Adonis analyzes and partitions the sum of squares using distance matrices. It can be seen as an ANOVA using distance matrices (analogous to MANOVA—multivariate analysis of variance). Therefore, it could test if two or more groups have similar compositions. Betadisper first calculates the average distance of group members to the group centroid in multivariate space (generated by a distance matrix). Then, an ANOVA is conducted to test if the dispersions (variances) of groups are different.

### 2.7. LASSO Method

In the LASSO algorithm, the feature importance was ranked in the framework of general linear models. Specifically, Tibshirani’s LASSO method was adopted (Tibshirani, 1996). LASSO arises from a constrained form of ordinary least squares regression, where the sum of the absolute values of the regression coefficients is constrained to be smaller than a specified parameter. The LASSO procedure offers extensive capabilities for customizing the selection with a wide variety of selection and stopping criteria, such as the Akaike information criterion (AIC).

### 2.8. Immunohistochemistry

The rats were euthanized and transcardially perfused with 25 mL of PBS and 4% paraformaldehyde. Whole brains were fixed with 4% paraformaldehyde for approximately 3 days. Paraformaldehyde-fixed 2 mm coronal slices were embedded in paraffin and cut into 5 µm thick sections, which were deparaffinized and rehydrated using a graded series of ethanol solutions. The sections then underwent an antigen retrieval process and were stained using horseradish peroxidase anti-myelin basic protein (MBP) antibody (BioLegend, catalog number 808405). Next, 3,3′-diaminobenzidine and hematoxylin staining was performed on sections by using the Chemicon IHC Select system (Millipore, catalog number DAB050). The sections were observed through microscopy (Olympus/Bx43). The MBP-positive area was calculated from sections by using HistoQuest tissue analysis software V2.0 (TissueGnostics, Vienna, Austria).

### 2.9. Statistical Analysis

Unpaired *t*-test were performed for behavioral data analyses using GraphPad Prism software. Error bars represent the standard error of mean. Microbiota enrichment analysis between the groups was conducted using the linear discriminant analysis (LDA) effect size (LEfSe) method. Data were compared using the Kruskal–Wallis and Wilcoxon tests; differences were considered significant at *p* ≤ 0.05 and a logarithmic LDA score of ≥ 2 [22]. Data were visualized as cladograms created using GraPhlAn [23].

## 3. Results

### 3.1. Maternal LPS Stimulation Causes Social Behavior Deficits and Anxiety-Like and Repetitive Behavior in MIA Male Offspring

No significant alterations in reproductive parameters were found in female rats exposed to prenatal LPS treatment (LPS-treated group) or PBS treatment (control group). The number of pups born in each litter, the parturition day, or individual bodyweight of offspring at 5 and 7 weeks showed no significant difference between prenatal LPS and PBS treatments (data not shown). To test whether MIA offspring began displaying social behavior deficits before or after sexual maturity, we examined their social behavior at 5 and 7 weeks. We used OptiMouse software to track the movement of rats in our three-chamber apparatus during the social behavior tests. Male offspring from the prenatal LPS-treated group spent less time in the social region of the central space in the three-chamber apparatus than those from the PBS-treated control group at both 5 and 7 weeks (Figure 1B,C), whereas female offspring had a normal social behavior (data not shown). Thus, male MIA offspring displayed social behavior deficits before and after sexual maturity. MIA male offspring also buried more marbles than the control male offspring (Figure 1D), which indicates anxiety-like and repetitive behavior. MIA and control male offspring had no differences in the total distance moved, number of entries, or time spent in the wall zone of the open-field assay (Figure 1E). Male offspring from the LPS group demonstrated a similar preference for the novel object compared with male offspring from the control group (Figure 1F). Taken together, the results revealed that locomotor activity and cognition were similar in LPS and control groups.

### 3.2. Fecal Microbiome Profile in Male MIA Offspring Is Similar to That of Patients with ASD

The gut microbiota profile in ASD was determined by identifying fecal microbiota through 16S rRNA gene sequencing and next-generation sequencing [24]. We observed that the fecal microbiota of prenatal LPS-treated male offspring had slightly, but not significantly, higher alpha diversity than that of the control offspring (Figure 2A). An unweighted and weighted UniFrac principal coordinate analysis indicated that the fecal microbiota profile of male MIA offspring was significantly different from that of PBS-treated male offspring (Figure 2B). A significant increase in *Fusobacteria* abundance and decrease in *Actinobacteria* abundance at the phylum level of microbial composition was observed in MIA offspring compared with that in control offspring (Figure 3 and Table 1). Compared with control offspring, MIA male offspring had a significantly increased abundance of *Fusobacteriaceae* and *Rikenellaceae* families (Figure 4A), and significantly decreased abundance of *Micrococcaceae*, *Staphylococcaceae*, *Aerococcaceae*, *Corynebacteriaceae*, and *Erysipelotrichaceae* families (Figure 4B). At the genus level, compared with control offspring, MIA male offspring had significantly increased *Ruminococcus_1*, *Fusobacterium*, *Acetatifactor*, *Alistipes*, and DNF00809 (Figure 4C), and significantly decreased *Coprococcus_3*, *Rothia*, *Sellimonas*, *Staphylococcus*, *Aerococcus*, *Corynebacterium_1*, *Candidatus_Stoquefichus*, and *Blautia* (Figure 4D).

### 3.3. Association of Fecal Microbiome Profile with Social Behavior and Anxiety-Like and Repetitive Behavior in MIA Male Offspring

The increased abundance of *Alistipes*, *Fusobacterium*, and *Ruminococcus* and decreased abundance of *Coprococcus* [25,26,27], *Erysipelotrichaies*, and *Actinobacteria* in LPS-induced MIA offspring were consistent with that of humans with ASD [28,29,30]. Next, we determined the association of ASD-related microbiota (increased or decreased amount of microbiota in the LPS group) with the level of the social behavior deficit and with anxiety-like and repetitive behavior by using the LASSO method. The LASSO procedure offers extensive capabilities to build a model that can determine the coefficient progression of selected microbiota with the profile of the indicated phenotypes (social behavior time and the percentage of buried marbles). The AIC method was applied to determine which microbiota profile was the best fit for the model that could reflect the profile of social behavior time and anxiety-like and repetitive behavior. Figure 5A indicates that the AIC values of *Alistipes* and phylum *Actinobacteria* were the smallest in terms of the information loss; thus, the abundance of *Alistipes* and phylum *Actinobacteria* had a higher association level with the profile of defective social behavior than the other microbiota, and *Fusobacterium* and *Coprococcus* had a higher association level with the profile of buried marble percentage based on AIC (Figure 5B).

### 3.4. Maternal LPS Stimulation Causes Hypomyelination in the Prefrontal Cortex and Thalamus Nucleus in MIA Male Offspring

Prenatal LPS stimulation at gestation days 15 and 16 causes abnormal myelination in the cortical and limbic brain regions in offspring [31]. In our study, prenatal LPS stimulation at gestation day 9.5 resulted in a significantly decreased MBP^+^ area in the prefrontal cortex (Figure 6A) and thalamic nucleus (Figure 6B) of the male MIA offspring (LPS group) than in control offspring by immunohistochemistry staining. In other words, maternal LPS stimulation caused hypomyelination in the prefrontal cortex and thalamic nucleus of adult male offspring.

## 4. Discussion

MIA causes an altered brain–gut–microbiota profile in male offspring, with autism-like phenotypes. In this study, we demonstrated that prenatal LPS stimulation caused altered social behavior and anxiety-like and repetitive behavior in male offspring, as revealed by our three-chamber test and marble-burying test, respectively. The NOR test and open-field assay demonstrated that MIA male offspring had a behavior profile similar to that of control offspring, indicating normal recognition memory and locomotor activity. MIA-induced hypomyelination in the prefrontal cortex and thalamic nucleus and altered microbiota profile provided evidence to show that LPS stimulation in the gestation stage causes altered brain–gut–microbiota axis phenotypes in offspring.

Patients with ASD have alterations in the gut microbiota composition compared with individuals without ASD [7,9]. In the present study, the gut microbiota diversity in male MIA offspring with ASD phenotypes was similar to the fecal microbiota profiles of children with ASD, such as a significant increase in *Ruminococcus*, *Fusobacterium*, and *Alistipes* [25,26,27]. We also observed a significant reduction in *Coprococcus*, *Erysipelotrichaies*, and *Actinobacteria*, which is consistent with the results of previous studies [28,29,30]. Compared with children without ASD, multiple microbiota species are elevated or reduced in children with ASD who have complex neurodevelopmental disorders involving disruptions in language and social behavior, restricted interests, and repetitive behaviors [9]. Microbiota transfer therapies have reported an improvement in ASD behavioral symptoms [32,33], suggesting that correcting the altered microbiota profile is a promising treatment strategy for ASD. Among the various changed microbiota species in MIA offspring, we demonstrated that the abundance of *Alistipes* and *Actinobacteria* was associated with the profile of social behavior, and *Fusobacterium* and *Coprococcus* was associated with the profile of anxiety-like and repetitive behavior; this suggests that *Alistipes, Actinobacteria, Fusobacterium,* and *Coprococcus* may be microbiome biomarkers or treatment targets for ASD.

MIA animal models are unique experimental tools for overcoming the limitations of epidemiological studies, such as the longitudinal evaluation of neurobiological processes from gestation to adulthood. Maternal exposure to LPS induces hypomyelination in the internal capsule and in newborn rats [34]. Maternal valproic acid stimulation causes hypomyelination of the prefrontal cortex and has been associated with social behavior deficit [35]. Poly I:C-induced MIA stimulation also causes a disruption in the myelin structure and a weakened thalamocortical connection in offspring [36]. In the present study, prenatal LPS-induced MIA caused the following changes in adult offspring: social behavior deficit, anxiety-like and repetitive behavior, ASD-related microbiota, and hypomyelination in the prefrontal cortex and thalamic nucleus. Prenatal LPS stimulation causes the inflammatory responses in brain of MIA offspring. The elevated levels of inflammatory cytokine (TNF-α and IL-1β) in the gestational stage are associated with behavioral impairment and hypomyelination [34,37]. The activation of an inflammatory reaction, including pro-inflammatory and anti-inflammatory cytokines, is associated with ASD [38,39,40,41]. In addition, poly I:C-induced MIA increases TNF-α and IL-1β expression in the colon of MIA offspring with an altered microbiota profile [42]. These results suggest that maternal LPS stimulation might impart an inflammatory reaction that is associated with an altered brain–gut–microbiota axis. Together, these findings suggest that MIA stress alters the brain–gut–microbiota axis leading to an inflammatory reaction, defective myelination and abnormal microbiota in offspring in ASD-related behavior deficits.

The most commonly used MIA animal models are prenatal poly I:C and LPS stimulation in pregnant animals. Poly I:C is a synthetic analogue of the viral double-strand RNA which activates Toll-like receptors (TLR)-3, whereas LPS is an endotoxin from Gram-negative bacteria which activates TLR-4 [43]. Prenatal poly I:C stimulation can alter the MIA microbiota profile in MIA offspring with a behavior impairment [29,42,44,45]. However, the link of an altered microbiota and social behavior impairment remains unclear in the LPS-induced MIA animal model. A previous study demonstrated several different effects of poly I:C and LPS prenatal stimulation on the behavior, development, and inflammatory response in pregnant mice and their offspring [46]. For example, prenatal LPS stimulation caused a decrease in the astrocytic marker (GFAP) and neuronal marker (NeuN) expression level in offspring at GD18, whereas GFAP and NeuN expression levels were not altered in the poly I:C-induced MIA model [46]. In addition, plasma cytokine IL-2, IL-5, IL-6, and inflammatory marker mGLuR5 were significantly increased in the brain of poly I:C-induced MIA offspring, but not in LPS-induced MIA offspring [46]. In this study, we found that human ASD-related *Ruminococcus* are increased in the LPS-induced MIA model, whereas *Ruminococcus* are decreased in the poly I:C-induced MIA model [29]. Taken together, different prenatal stimulation sources (virus or bacteria) may induce a differential profile of brain cell markers and microbiota profile in MIA offspring.

The prevalence of ASD is higher in males, with an approximate male-to-female ratio of 3:1 [47]. In this study, our MIA female offspring had normal social behavior when compared to the control group. In addition, sex may affect the microbiota profile [48]. For example, *Coprococcus* was less present in the microbiota profile of male mice compared to female mice [49]. These data were consistent to our study, revealing a decreased number of *Coprococcus* in the microbiota profile of male rats compared to female rats (data not shown). A human study showed that healthy males possess higher levels of *Fusobacterium* than females [50]. In an ASD predominantly male cohort study (18 males and 2 females), ASD children had decreased *Coprococcus* than the healthy subjects [28]. Another study showed that ASD patients with constipation (25 male and 5 female subjects) had increased *Fusobacterium* compared with healthy subjects [26]. Taken together, this evidence suggests that the differences in microbiota profiles between male and female subjects may be correlated to the altered microbiota profiles of ASD patients.

## 5. Conclusions

In the present study, LPS-induced MIA offspring displayed altered brain–gut–microbiota axis phenotypes, including social behavior deficits, anxiety-like and repetitive behavior, a human ASD-like microbiota profile (higher abundance of *Alistipes*, *Fusobacterium*, and *Ruminococcus* and a lower abundance of *Coprococcus*, *Erysipelotrichaies*, and *Actinobacteria* than control offspring), and hypomyelination in the prefrontal cortex and thalamic nucleus. An abundance of *Alistipes* and *Actinobacteria* was the most relevant for the profile of defective social behavior, whereas the abundance of *Fusobacterium* and *Coprococcus* was associated with anxiety-like and repetitive behavior. These potential ASD-related microbiomes require further studies to prove their direct association to ASD-related behavior, such as fecal transplantation or target microbiome transplantation therapy. The relative mechanisms of ASD etiopathogenesis still remain unknown and further studies are needed to prove the cause–effect relations between microbiome dysbiosis and ASD. Our findings provide insights into the relationship between maternal infection and the etiopathogenesis of ASD with an abnormal brain–gut–microbiota axis.

## Figures and Tables

**Figure 1 brainsci-11-01085-f001:**
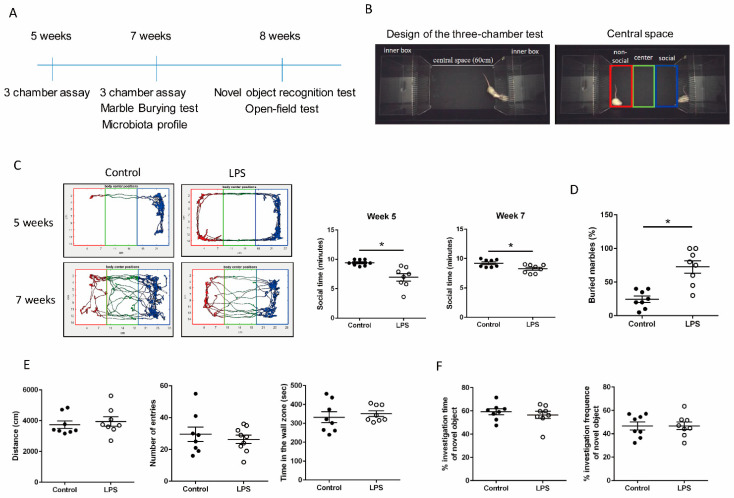
Maternal lipopolysaccharide (LPS) stimulation caused social behavior deficits and anxiety-like and repetitive behavior in male offspring. (**A**) Timeline for behavioral experiments. The age of the rats in weeks is indicated on the timeline. (**B**) Design of the three-chamber test apparatus for rats. The apparatus contained three parts: two side chambers and one central space. The length of the central space was 60 cm. The right-side chamber housed the stranger rat. The test rat was placed in the central space and allowed to freely interact with the stranger rat. The central space was divided into three regions: social, center, and nonsocial. (**C**) Quantification of time spent in the social region. The social behavior of 5- and 7-week-old maternal immune activation (MIA) and control offspring was detected using a three-chamber apparatus. Their tracks in the central space of the three-chamber apparatus are indicated, with the nonsocial region indicated in red, the center region in green, and the social region in blue. * *p* < 0.05 (*n* = 8 per group). (**D**) Percentage of buried marbles in the MIA and control offspring. * *p* < 0.05 (*n* = 8 per group) (**E**) Locomotor activity evaluation was performed in 7-week-old MIA and control male offspring using the open-field test (*n* = 8 per group). (**F**) The NOR test was performed on 7-week-old MIA and control male offspring (*n* = 8 per group). All data are presented as mean ± SEM.

**Figure 2 brainsci-11-01085-f002:**
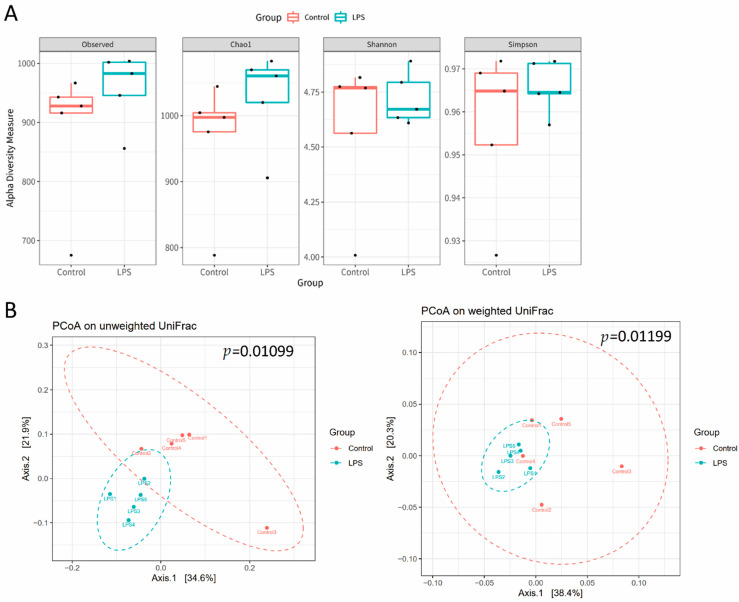
Fecal microbiome distribution in male maternal immune activation (MIA) offspring. The feces of prenatal lipopolysaccharide (LPS)-stimulated male offspring (LPS group) and control rats were prepared for fecal microbiome profiling through high-throughput sequencing of the 16s rRNA gene on the Illumina MiSeq system. (**A**) Alpha diversity of MIA offspring (LPS) and controls. (**B**) Principal coordinate analysis plot based on unweighted or weighted UniFrac distance of omeprazole MIA offspring and control samples. A significant difference in beta diversity was evaluated using permutational multivariate analysis of variance (vegan:adonis, 1000 permutations), and beta dispersion was quantified using a betadisper (vegan:betadisper, 1000 permutations). Both indices achieved adonis *p* < 0.05 and betadisper *p* > 0.05.

**Figure 3 brainsci-11-01085-f003:**
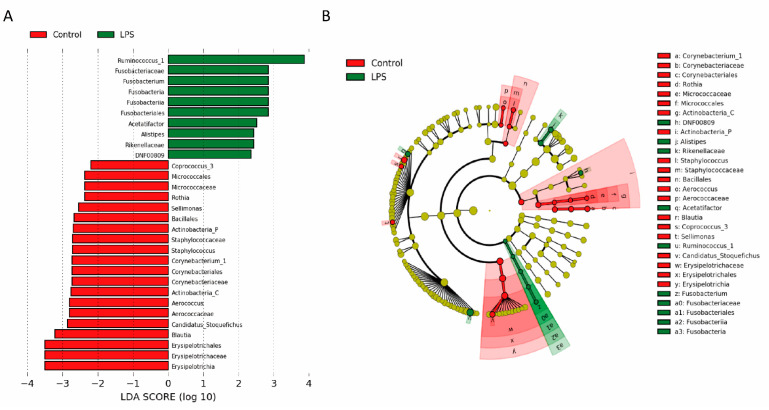
Fecal microbiota was changed in male MIA offspring. (**A**) Linear discriminant analysis (LDA) effect size analysis of gut microbiota changed in MIA offspring (lipopolysaccharide (LPS)) and control rats. Significant biomarkers were defined as taxa with an LDA score (log10) of ≥2. (**B**) Significant taxa differences in MIA offspring (LPS) are highlighted on the cladogram.

**Figure 4 brainsci-11-01085-f004:**
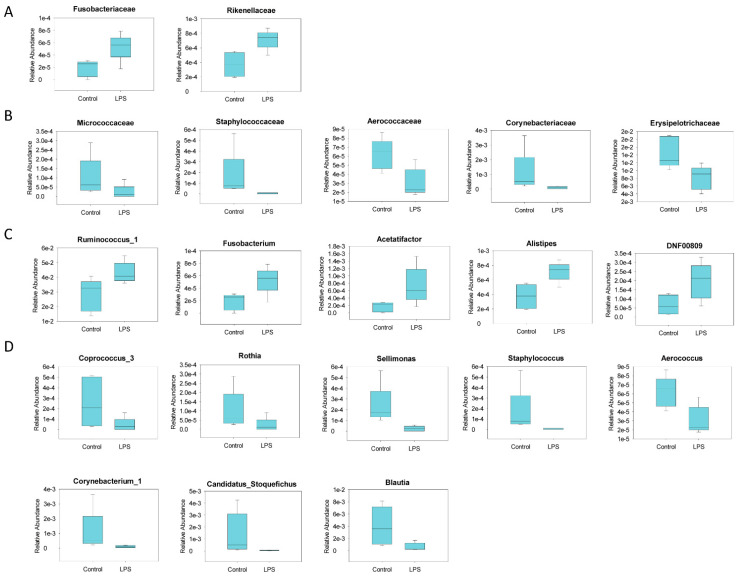
Fecal microbiota at the family and genus levels with significant changes in abundance in maternal immune activation (MIA) offspring. A significant (**A**) increase and (**B**) decrease in bacteria at the family level in MIA offspring (lipopol ysaccharide (LPS)) compared with control rats. Similarly, significant increase (**C**) and decrease (**D**) in bacteria at the genus level were observed in MIA offspring (LPS) compared with control rats. *n* = 5 per group. All data had LDA scores ≥ 2. Relative abundance indicates number of reads of targeted microbes per sample/total number of reads per sample.

**Figure 5 brainsci-11-01085-f005:**
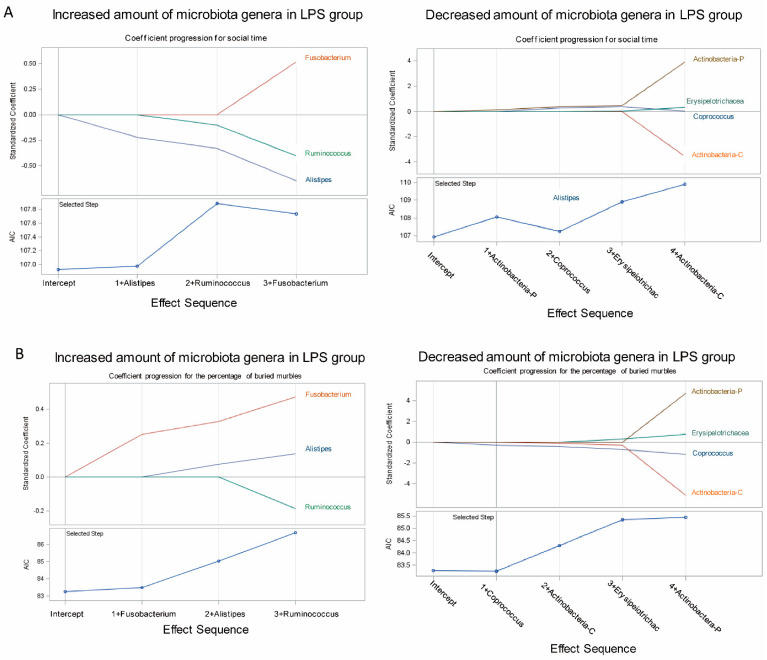
Feature importance of bacterial species for social behavior deficit and anxiety-like and repetitive behavior. (**A**) Standardized coefficients of the effects selected (the increased or decreased amount of microbiota genera in the lipopolysaccharide (LPS) group) with social time or (**B**) the percentage of buried marble at a given step of the stepwise method are plotted as a function of the step number. The coefficients plot displays the values of the estimates for each model at each iteration step. The effect was added into the model in an order of its relative importance measured with AIC. The plot labels the added effects at each step. The vertical axis of plot shows standardized estimates that can track the change of AIC for each successive model. Each colored line visualizes the evolution of values for a particular effect. The AIC plots show the relative importance of the effects selected (microbiota species) at steps of the selection process when effects entered the model. Actinobactera-P indicates the relative amount of phylum *Actinobacteria*. Actinobacteria-C indicates the relative amount of class *Actinobacteria*.

**Figure 6 brainsci-11-01085-f006:**
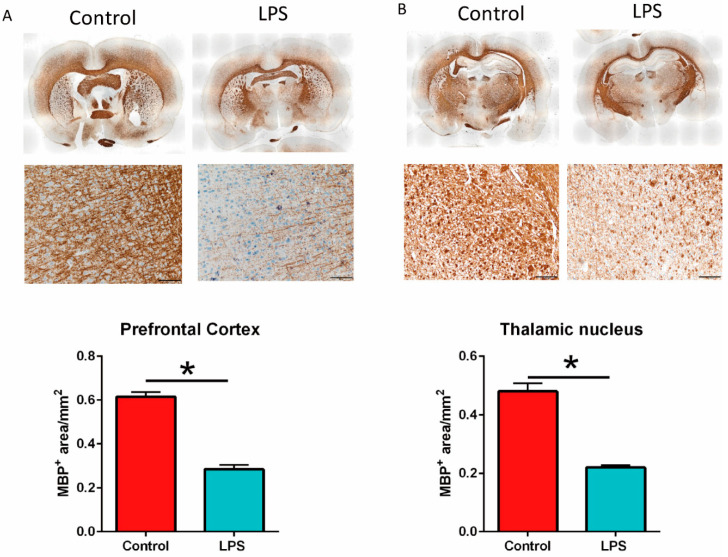
Myelin basic protein (MBP) expression level in the prefrontal cortex and thalamic nucleus in maternal immune activation (MIA) male offspring. (**A**) The MBP expression level in the prefrontal cortex and (**B**) thalamic nucleus of control and MIA offspring (lipopolysaccharide—(LPS) group) was detected with immunohistochemistry staining. The MBP expression level in whole brain slices are shown in the upper pictures. The MBP expression level in prefrontal cortex and thalamic nucleus are shown in the lower pictures. The quantification of the MBP^+^ area in the prefrontal cortex and thalamic nucleus is shown in the bar graph. * *p* < 0.05 (*n* = 4 per group). All data are presented as mean ± SEM. Scale bar, 100 μm.

**Table 1 brainsci-11-01085-t001:** Phylum-level microbiome composition in the feces of control and maternal immune activation male offspring. The relative abundance of microbiomes in phylum level from the feces of control and LPS MIA offspring were obtained from 16S rRNA gene sequencing data. The significant difference of group was statistically calculated by U-test. SD: standard deviation.

Phylum	Control		LPS		
	Mean	SD	Mean	SD	U-Test
Actinobacteria	1.7 × 10^−3^	1.3 × 10^−3^	8 × 10^−4^	1.0 × 10^−4^	0.01
Bacteroidetes	5.8 × 10^−1^	4.0 × 10^−2^	6 × 10^−1^	1.0 × 10^−2^	0.2
Deferribacteres	2.0 × 10^−4^	2.0 × 10^−4^	4 × 10^−4^	1.0 × 10^−4^	0.2
Firmicutes	3.8 × 10^−1^	3.4 × 10^−2^	3.6 × 10^−1^	1.0 × 10^−2^	0.149
Fusobacteria	1.8 × 10^−5^	1.3 × 10^−5^	5.3 × 10^−5^	2.2 × 10^−5^	0.03
Patescibacteria	4.4 × 10^−3^	4.8 × 10^−3^	5.7 × 10^−3^	3.7 × 10^−3^	0.037
Proteobacteria	1.1 × 10^−2^	3.7 × 10^−3^	8.2 × 10^−3^	4.4 × 10^−3^	0.105
Tenericutes	8.0 × 10^−4^	1.8 × 10^−3^	4.2 × 10^−3^	9.5 × 10^−3^	0.416
Verrucomicrobia	5.2 × 10^−3^	1.1 × 10^−2^	6.7 × 10^−6^	1.0 × 10^−5^	0.086

## Data Availability

The datasets used and/or analyzed during the current study are available from the corresponding author on reasonable request.

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
