# Peer review of "Maternal Immune Activation Causes Social Behavior Deficits and Hypomyelination in Male Rat Offspring with an Autism-Like Microbiota Profile"

_brainsci, 2021, doi:10.3390/brainsci11081085_

Round 1

Reviewer 1 Report

Very interesting research article.

I found it well planned and performed, results are clearly showed and discussion is consistent.

Another main finding of this research is the hypomyelination in the brain areas of ASD-like animals. Also considering the LPS is an external chemical and that it affects the immune system cells, authors could add a little bit more in discussion section on the interplay between immune and gut-brain-axis in ASD.

if authors could more discuss about the interplay between immune and gut-brain-axis in ASD.

Authors used LPS as triggering molecule for ASD model, LPS affects immune system cells, in this way there is a contribution by an altered immune system/gut-brain-axis in ASD pathogenesis.

Reviewer 2 Report

The study entitled “Maternal immune activation causes social behavior deficits and hypomyelination in male rat offspring with an autism-like microbiota profile” provide experimental evidence of maternal immune activation when the maternal immune system is triggered by an infectious agent, in this study the authors injected the pregnant rats intraperitoneally with LPS and the litters after weaning from postnatal 21 days, separately caged the pups based on sex and performed several behavioral tests in male pups, to access their ability and compared the outcome with the control pups (liters from PBS injected pregnant rats). Additionally, they performed several behavioral tests and the microbiota was assessed in the feces to show a difference, which is attributed to ASD-like profile, concluding that there were increased levels of Ruminococcus, Fusobacterium, and Alistipes and reduced levels of Coprococcus, Erysipelotrichaies, and Actinobacteria. They inferred that an abundance of Alistipes and Actinobacteria being associated with social behavior, and Fusobacterium and Coprococcus are associated with anxiety-like and repetitive behavior, suggesting, that Alistipes, Actinobacteria, Fusobacterium, and Coprococcus could be microbiome biomarkers or treatment targets for ASD.

Comments and suggested revision to improve the manuscript:

  1. Was the LPS injected intraperitoneal dose reported correct? Is it 500 micrograms or grams? The dosage may be too low to elicit such a profound immune response. Please check the dose. Were the pregnant rats injected once or more times?
  2. Provide catalog numbers for horseradish peroxidase anti–myelin basic protein (MBP) antibody (Abcam) and Chemicon IHC Select system (Millipore)
  3. Figure 1: the dot plots showing social time, it would be meaningful for visual comparison to have the y-axis same scale between week 5 and 7 and preferably the scale in minutes rather than seconds. In Week 7, the disconnected Y-axis does not make sense. Please revise the figure.
  4. Figure 4: correct the Y-axis label to Relative Abundance. How was this abundance estimated? As previously mentioned, if the Y-axis scale is comparatively maintained constant, the difference and significance can be easily visualized. Provide some details in the figure legend. I assume this is one of the significant findings and having experimental evidence with a large set of data it would be valuable to provide details on how many pups were included in each group and describe whether they were all fed the same diet and if you have weighed the pups provide those details as well.
  5. Figure 5: Provide a more description in the figure legend. Was this in comparison with the control pups? Without a clear description, this figure will not be appreciated by the readers. Describe in the legend what the Y-axis means and does the x-axis refer to the progression in the colonization of the different microbiota?
  6. Table 1: Please describe in the figure legend how were these values obtained. Were these values obtained from the RNA sequence?
  7. The authors have not provided any details on the RNA sequencing data, it would be valuable to mention the outcome of those experiments.

Minor corrections:

Pg3 Ln110: three-chamber apparatus

Pg3 Ln131: evaluate the general

Pg4 Ln 190: t-test

Pg5 Ln202: offspring at 5 and 7 weeks were no significant difference between

Change to ‘offspring at 5 and 7 weeks did not show a significant difference between’ of ‘offspring at 5 and 7 weeks were not significantly different between

Pg9 Ln274: A significant increase (A)

Reviewer 3 Report

In ther manuscript, Lee and co-authors  studied lipopolysaccharide (LPS)-induced maternal immune activation (MIA) offspring and found it to have an abnormal brain–gut–microbiota axis with social behavior deficits, anxiety-like and repetitive behavior, hypomyelination, and an ASD-like microbiota profile.

1) The deleterious impact of LPS is well-known, but the manuscript declares to present the first ever attempt to link the LPS-induced MIA to gut microbiota shifts. However, the authors’ declaration that (I quote) “alterations in the brain–gut–microbiota axis in lipopolysaccharide (LPS)-induced MIA offspring remains unclear” has appeared contradictory to the article database search results. So, a brief Pubmed inquiry yielded recently published reports of changes in gut microbiota composition alongside with behavioral disturbances in ferret (PMID 30406186), mice (PMID 2946211) and rat (PMID 33793085) animal models. Thus, the first and major concern is the manuscript’s scientific novelty and originality.

Other, minor concerns about the manuscript are the following:

2) English grammar should be checked by a professional translator. For example, lines 32-33: “…the alterations in the brain–gut–microbiota axis in lipopolysaccharide (LPS)-induced MIA offspring remains unclear” should be corrected for “…remain unclear”. Line 201-202: “The number of pups born in each litter, the parturition day, or individual bodyweight of offspring at 5 and 7 weeks were no significantly difference between…” should be replaced with “…showed no significantly difference between…” or “…demonstrated no significantly difference between…”.

3) Lines 262-264: what are “vegan”, “adonis”, “betadisper”? These terms should be explained.

4) In discussion, the authors are recommended to propose a mechanism and/or quote hypotheses of other researchers to explain how MIA affects the offspring’s gut microbiota composition and how it further influences the nervious system and behavior; whether the gut microbiota changes are causative for the ASD-like signs or just indicators of some underpinning processes; and why the impact is sex-specific, i.e. why the affected animals are males only. Of course, the article does not address these questions directly, however, a brief attempt to describe the possible cause-effect relations would greatly improve the manuscript.
